# Antimicrobial Resistance in Rapidly Growing Nontuberculous Mycobacteria among Domestic and Wild Animals Emphasizing the Zoonotic Potential

**DOI:** 10.3390/microorganisms11102520

**Published:** 2023-10-09

**Authors:** Irena Reil, Silvio Špičić, Ljubo Barbić, Sanja Duvnjak, Gordan Kompes, Miroslav Benić, Dora Stojević, Željko Cvetnić, Jurica Arapović, Maja Zdelar-Tuk

**Affiliations:** 1Croatian Veterinary Institute, 10000 Zagreb, Croatia; irena.reil@gmail.com (I.R.); spicic@veinst.hr (S.Š.); benic@veinst.hr (M.B.); dstojevic@gmail.com (D.S.); cvetnic@veinst.hr (Ž.C.); zdelar-tuk@veinst.hr (M.Z.-T.); 2The Faculty of Veterinary Medicine, University of Zagreb, 10000 Zagreb, Croatia; ljbarbic@gmail.com; 3Department of Infectious Diseases, University Clinical Hospital Mostar, 88000 Mostar, Bosnia and Herzegovina; jurica.arapovic@mef.sum.ba; 4School of Medicine, University of Mostar, 88000 Mostar, Bosnia and Herzegovina

**Keywords:** non-tuberculous mycobacteria, rapidly growing non-tuberculous mycobacteria, antimicrobial susceptibility testing, antimicrobial resistance, zoonosis

## Abstract

Non-tuberculous mycobacteria (NTM) are opportunistic pathogens capable of causing infections in humans and animals. The aim of this study was to demonstrate the potential role of domestic and wild animals as a reservoir of multiple resistant, rapidly growing NTM strains representing a potential zoonotic threat to humans. A total of 87 animal isolates belonging to 11 rapidly growing species (visible colonies appear within three to seven days) were genotyped and tested for susceptibility to the 15 most commonly used antibiotics in the treatment of such infections in a human clinic. By determining the antimicrobial susceptibility, the most prevalent resistance was found to cephalosporins (>50%), followed by amoxicillin–clavulanate (31.0%), clarithromycin (23.0%), tobramycin (14.9%) and doxycycline (10.3%). Resistance to imipenem, ciprofloxacin, minocycline and linezolid was notably lower (<7.0%). All tested isolates were susceptible to amikacin and moxifloxacin. The most frequent resistance was proved in the most pathogenic species: *M. fortuitum*, *M. neoaurum*, *M. vaccae* and *M. porcinum*. Meanwhile, other species displayed a higher sensitivity rate. No significant resistance differences between domestic and wild animals were found. The established significant frequency of resistance highlights the significant zoonotic potential posed by circulating rapidly growing NTM strains, which could lead to challenges in the treatment of these infections.

## 1. Introduction

The genus *Mycobacterium* includes around 200 species that differ in terms of metabolism, growth rate, epidemiology, pathogenicity, geographical distribution and antimicrobial susceptibility [1]. In addition to species that cause tuberculosis (*Mycobacterium tuberculosis* complex), there are also species commonly referred to as non-tuberculous mycobacteria (NTM) that act as opportunistic pathogens capable of causing infections in humans and animals [2]. Often termed ecological mycobacteria, NTM species have been isolated from water, soil, dust and plants [3]. According to cultural characteristics, they are categorized as rapidly growing and slow-growing species [4]. Among rapidly growing mycobacteria (RGM), there are more than 75 different species, representing about 50% of all recognized mycobacterial species [5].

One of the characteristics of NTM species is a high level of natural drug resistance, which does not have to be specifically reflected in the minimum inhibitory concentration (MIC) values of a certain tested antibiotic, and this is the most disturbing and contradictory feature of NTM lung diseases. The best-known example of this phenomenon is the inducible macrolide resistance gene or erythromycin resistance methylase (*erm*) gene, whose activity can only be detected in vitro by prolonged incubation of the strain in the presence of macrolides [6,7]. Unfortunately, in addition to natural drug resistance, many NTM pathogens are also subject to inducible and mutational resistance acquired during suboptimal exposure and drug selection [6]. Finally, the future of treatment of diseases caused by NTMs should focus on identifying innate mechanisms of antibiotic resistance and discovering ways to overcome them. Furthermore, mobile genetic elements associated with antimicrobial resistance in bacteria have recently been described, through which mycobacteria acquire the ability to become immediately resistant. This fact also represents a significant evolutionary adaptation [8]. For this reason, their management poses greater challenges than the treatment of classical tuberculosis, and so far, no standardized antimicrobial therapy for NTM has been established. Different types of NTM have different profiles of antimicrobial susceptibility, but there are few publications that indicate those differences [9]. Current antimicrobial regimens for the treatment of RGM diseases are based on their unique in vitro susceptibility patterns, and antibiotics of different classes must be included in the therapy due to the fact that there are multidrug-resistant strains among NTMs [9,10]. There is no question that there are many weaknesses and gaps in our knowledge about diseases caused by NTM.

In recent years, human NTM infections have significantly increased [11]. RGM-caused infections are reported predominantly from developed regions of the world, manifesting as skin, soft tissue, bone and pulmonary disease [4]. The concerning aspect is that the reservoir for RGM pulmonary disease is still unknown, and the transmission routes of most infections have not been explained so far [4,12]. With an increase in the average age of the human population and a higher percentage of immunosuppressed people, we can expect that the prevalence of NTM infections will continue to increase [13].

According to our knowledge and available data, in the field of veterinary medicine, there are only a few studies so far that describe antimicrobial susceptibility testing (AST) of NTM isolates originating from animals. There is also a lack of research focused on species-specific environmental niches associated with human infection that could clarify the pathways and mechanisms of transmission of such infections. Therefore, we performed AST on rapidly growing NTM species originating from domestic and wild animals using the most commonly used antibiotics for the treatment of NTM human infections to assess potential zoonotic implications and to understand the potential role of animals as a reservoir of multiple resistant RGM strains.

## 2. Materials and Methods

### 2.1. Bacterial Strains

The isolates originating from domestic and wild animals were collected in the Laboratory for Bacterial Zoonoses and Molecular Diagnostics of Bacterial Diseases (National Reference Laboratory for bovine tuberculosis) at the Croatian Veterinary Institute as a part of bovine tuberculosis eradication and surveillance programs prescribed by the Croatian Ministry of Agriculture in the period from year 2012 to 2015. Samples were obtained when there was a positive tuberculin test reaction or observable pathological changes suggesting tuberculosis during slaughter in domestic animals and after a random shooting during the implementation of the above-mentioned program in wild animals. The research covered the area of 14 counties and the City of Zagreb within the Republic of Croatia.

Identification of mycobacterial isolates to the species level was performed by the GenoType Mycobacterium CM/AS line probe assay (Hain Lifescience, Nehren, Germany), as well as sequencing of 16S rRNA [14], *rpoB* [15], *hsp*65 parts of genes [15,16], and the ITS region [17] for those unidentified by the previously mentioned method. After identification, the strains were stored at −80 °C until the AST was carried out.

### 2.2. Antimicrobial Agents

Susceptibility testing was performed on the 15 most commonly used antibiotics in the treatment of NTM diseases caused by rapidly growing species in a human clinic: amikacin (AMI), amoxicillin–clavulanate (AUG2), cefepime (FEP), cefoxitin (FOX), ceftriaxone (AXO), ciprofloxacin (CIP), clarithromycin (CLA), doxycycline (DOX), imipenem (IMI), linezolid (LZD), minocycline (MIN), moxifloxacin (MXF), tigecycline (TGC), tobramycin (TOB) and trimethoprim-sulfamethoxazole (SXT).

### 2.3. Antimicrobial Susceptibility Testing (AST)

Testing was performed by standard method for determining antimicrobial resistance of mycobacteria using Thermo Scientific ™ Sensititre ™ Myco RAPMYCO AST Plate commercial kit ((Thermo Fisher Scientific, Waltham, MA, USA) for all isolates as recommended by the Clinical and Laboratory Standards Institute [18,19]. Bacterial suspensions were prepared using cation-adjusted Mueller–Hinton broth according to the manufacturer’s instructions (Thermo Fisher Scientific) and incubated at 30 ± 2 °C. We first inspected the test plates after 48 h. The assessment of growth inhibition after 48 h is particularly useful for the *M. fortuitum* group and *M. mucogenicum* group, which is explained in more detail later. If the growth in the control well was sufficient, we recorded the minimum inhibitory concentration (MIC) values. Otherwise, we read it on the third or fourth day of incubation. The final reading should be at no more than five days, except for CLA. While examining the members of *M. phocaicum* and *M. peregrinum*, the test results of the CLA sensitivity (interpretation only, without MIC) were read with the other results after three to five days of incubation, since the mentioned species have non-functional or absent erythromycin resistance methylase (*erm*) gene responsible for macrolide resistance and therefore expected to be susceptible to clarithromycin [7,20,21,22]. For every other tested species, we read the results for CLA on the 14th day of incubation, for the purpose of phenotypic detection of inducible resistance to macrolides caused by the presence of the *erm* gene unless the MIC value was ≥16 µg/mL earlier. Furthermore, when reading sensitivity to IMI in members of the *M. fortuitum* group (*M. fortuitum*, *M. peregrinum*, *M. septicum*, *M. porcinum*) and *M. mucogenicum* groups (*M. phocaicum*) in case the MIC value was >8 µg/mL on the fifth day of reading, the test was repeated, with incubation limited to three days. If the value of MIC of the repeated test was again > 8 µg/mL, we did not report the MIC value due to antibiotic instability. For TGC, we recorded only MIC values without results interpretation because breakpoint concentrations have not yet been established. For all other tested antimicrobials, we recorded MIC values and interpreted the results according to CLSI guidelines [18,19], as shown in Table 1. MIC was the lowest concentration of antimicrobial substance that inhibits >99% of mycobacterial growth for all antimicrobials, except SXT, for which MIC was around 80% of growth inhibition. For FEP, AXO, AUG2, and MIN guidelines for the results interpretation of rapidly growing NTM have not yet been established by CLSI, so we used CLSI guidelines for testing *Nocardia* spp. and other *Actinomycetes* [19,23] as described in a recent study [24]. Reference strain *M. smegmatis* ATCC 19420 (American Type Culture Collection, Manassas, VA, USA) was used as a quality control strain in the antimicrobial susceptibility tests.

### 2.4. Statistical Analysis

The statistical program Stata 13.1 (Stata Corp., College Station, TX, USA) was used to process the results. We presented the descriptive data as a total number and percentage values. The chi-square test was used for testing the observed differences between groups. Differences were deemed statistically significant with a *p*-value less than 0.05.

## 3. Results

We analyzed a total of 87 bacterial isolates belonging to 11 different rapidly growing NTM species, namely *M. arupense* (2 isolates), *M. chitae* (6 isolates), *M. elephantis* (1 isolate), *M. fortuitum* (27 isolates), *M. neoaurum* (25 isolates), *M. peregrinum* (1 isolate), *M. phocaicum* (7 isolates), *M. porcinum* (2 isolates), *M. pulveris* (1 isolate), *M. septicum* (1 isolate) and *M. vaccae* (14 isolates). Out of the total number of isolates, 15 came from domestic (12 cattle, 1 sheep, 1 pig, 1 chicken) and 72 from wild animals (40 roe deer, 32 wild boars).

We detected resistance to most of the antimicrobials across nearly all tested rapidly growing NTM species. Among the *M. fortuitum* isolates, 81.5% were resistant to AXO, while 77.8% were resistant to FEP and 66.7% to AUG2. Likewise, 68% of *M. neoaurum* isolates were resistant to FEP, while 64.3% of *M. vaccae* isolates were resistant to AXO and 50% to FEP. Among two tested *M. porcinum* isolates, 100% were resistant to AXO, FEP, CLA, DOX and TOB, while 50% were resistant to AUG2 and MIN. *M. peregrinum* isolate was resistant to AUG2, FEP, AXO and TOB, while *M. elephantis* isolate was resistant to IMI. Isolates of *M. arupense*, *M. pulveris* and *M. septicum* were sensitive to all tested antimicrobials. The prevalence of resistance to each antibiotic among the 11 rapidly growing NTM species is shown in Table 2, and more detailed results, together with MICs, are reported in Appendix A.

The antibiotic class against which resistance was most prevalent among tested isolates was cephalosporins (FEP and AXO, while resistance to FOX was less prevalent), followed by AUG2, CLA, TOB and DOX. Resistance to IMI, CIP, MIN and LZD was substantially less prevalent. All 87 isolates were susceptible to AMI and MXF (Table 3).

Of the 87 isolates tested, 32 (36.8%) were multidrug-resistant, which indicates resistance to two or more classes of antibiotics. (Table 4).

The highest percentage of multiple resistant strains according to the antibiotics used (at least four antibiotics) was observed in *M. fortuitum* species. There were statistically significant differences in antibiotic sensitivity among the species of rapidly growing mycobacteria (*p* < 0.001) (Table 5).

## 4. Discussion

The highest rate of resistance among our isolates was found in the group of cephalosporins, mostly cefepime and cefriaxone, while a much smaller percentage of resistance was found in cefoxitin. From the previous studies, it is evident that cefoxitin was the only cephalosporin included in the majority of treatment regimens for such infections in human medicine, and the drug is also one of the recommended for the treatment of the most common rapidly growing NTM infection in humans [9]. Similarly, we detected a high prevalence of resistance to amoxicillin–clavulanate and clarithromycin, followed by tobramycin, doxycycline and minocycline. In addition, we found a few isolates to be resistant to imipenem, linezolid, trimethoprim/sulfamethoxazole and ciprofloxacin. Of the above, clarithromycin, imipenem, linezolid, trimethoprim/sulfamethoxazole, tobramycin, doxycycline and ciprofloxacin are also among the recommended for the treatment protocol of the most common rapidly growing NTM infection in humans [9], and according to the previous reports, clarithromycin is the first drug of choice in every successful treatment regimen, which is stated below in text.

There is not enough information in the literature describing the AST of rapidly growing NTM strains isolated from animals. Therefore, we compared our results with those available in the literature on human isolates worldwide to see the zoonotic potential, as well as the role of animals as a reservoir of multidrug-resistant NTM strains.

Among *M. arupense* isolates, we did not detect resistance to any tested antibiotic. In humans, this species has been described so far as a cause of disseminated disease in immunocompromised patients [25], osteoarthritis [26], lung infections [27], and most often, tenosynovitis [28,29].

Similarly, among *M. pulveris* and *M. septicum* isolates, we did not detect resistance to any tested antibiotic. According to our knowledge, no infection caused by *M. pulveris* has been described in human medicine so far, while those caused by *M. septicum* were described in both healthy and immunocompromised individuals, but optimal therapeutic regimens have not been established [30].

There are very few descriptions of infections caused by *M. chitae* in human medicine; only pulmonary infections have been reported so far, while the treatment has not been described [31]. Our *M. chitae* isolates exhibited resistance to amoxicillin–clavulanate, clarithromycin and cefepime.

Our *M. elephantis* isolate from cattle showed resistance to imipenem. In humans, there are also very few descriptions of infections with this NTM species; it is known to cause respiratory infections and enteritis. Although the species *M. elephantis* was first described in an elephant that was thought to be a reservoir [32], the patient generally had no contact with animals [33].

The species *M. fortuitum* is one of the most frequently represented species of NTM in human patients in Europe and is considered one of the most important species globally [34]. Most often, it causes infections of the skin, soft tissues and bones [35], but lung infections have also been described [36]. The recommended antimicrobial drugs for the treatment of *M. fortuitum* infections with the percentage of sensitivity of the strains are amikacin (sensitivity 100%), ciprofloxacin (100%), sulfonamides (100%), cefoxitin (50%), imipenem (100%), clarithromycin (80%) and doxycycline (50%), but the recommendation is to determine the antimicrobial sensitivity of each isolated strain [9,37]. Of the mentioned antibiotics, in our research, we proved resistance to cefoxitin, doxycycline and clarithromycin among our isolates.

One of the recommended antimicrobials for treating infections caused by *M. neoaurum* in human medicine are linezolid, trimethoprim-sulfamethoxazole, clarithromycin, tobramycin and cefoxitin [9], while we described resistance exactly to these antibiotics among our isolates. *M. neoaurum* in human medicine has been described as the cause of bacteremia caused by a contaminated catheter, which is the most common pathology caused by this species, where the isolate proved to be resistant only to clarithromycin [38]. Meningoencephalitis [39] and skin infection in an immunocompetent patient [40] caused by this NTM have also been described, as well as a lung infection where the authors describe the sensitivity of the isolate to all tested antibiotics [41].

There are not many descriptions of infection caused by the species *M. peregrinum* in human medicine, although they account for approximately 2% of RGM infections [5]. Lung infections caused by *M. peregrinum* have also been described, but there is currently no established treatment for such infection [42]. Similarly, *M. phocaicum* was described as a cause of lung infection and was simultaneously isolated from the pool used by the patient [43]. Among our *M. peregrinum* and *M. phocaicum* isolates, we detected resistance to tobramycin, amoxicillin–clavulanate, cefriaxone and cefepime.

Infections caused by *M. porcinum* are an increasingly common finding in human medicine [44]. Wound infections, catheter-induced bacteremias, and rarely, lung infections caused by this NTM have been described, and the sensitivity of isolates to clarithromycin has been recorded [45]. In our isolates, we detected resistance to clarithromycin.

We detected resistance to doxycycline and ciprofloxacin among the *M. vaccae* isolates, which contrasts with the current use of those antibiotics in the treatment of infections in humans. The species *M. vaccae* in human medicine has so far been described as weakly pathogenic, although skin and lung infections have been described [46]. On the other hand, some studies indicate that exposure to *M. vaccae* has a positive effect on the psychological state of man [47].

From the above, we can conclude that the rapidly growing NTM isolates described in our study showed certain percentages of resistance to most of the antibiotics recommended for humans. We should not ignore the fact that infections caused by rapidly growing NTM species are very difficult to treat, with a high percentage of recurrence [4]. Antibiotics are the basis of treatment for NTM infections; however, each NTM species and each patient in human medicine requires different combinations of antibiotics, and the use of in vitro susceptibility tests is very limited, leading to difficulties in treatment [48]. The basis of all currently described treatment protocols for NTM infections are macrolides [4], while in our research we detected 23% of strains resistant to the macrolide clarithromycin. Furthermore, we detected the highest percentage of multi-resistant isolates (to four or more antibiotics) according to the antibiotics used in *M. fortuitum* and *M. neoaurum* isolates, which are considered to be among the most pathogenic species [5]. It is also important to note that we detected 11 multidrug-resistant isolates that showed resistance to five or more antibiotics among *M. fortuitum*, *M. neoaurum*, *M. porcinum* and *M. vaccae*, of which two isolates were isolated from domestic animals and nine from wild animals.

In order to clarify the role of domestic and wild animals in the spread of multidrug-resistant NTM strains, the following facts should be taken into account. By destroying and encroaching on natural habitats, wild animals are forced to approach urban areas, thus coming into direct or indirect contact with domestic animals. In such circumstances, the transfer of resistant strains of NTM from domestic to wild animals can occur, followed by spreading to a much wider area and transfer to other pastures and farms. In our research, we did not prove the differences in the resistance of NTM strains between wild and domestic animals, although it can be assumed that strains from domestic animals would show greater resistance due to frequent exposure to antibiotics. The data on the high percentage of multidrug-resistant strains isolated from wild animals, which we consider to be wild isolates that have likely not been exposed to antibiotics, is worrying. In this case, the question arises as to what extent the environment is contaminated by multidrug-resistant, rapidly growing NTM strains. For this reason, there is a need for further research into ecological niches in order to clarify the diseases caused by rapidly growing NTM species.

## 5. Conclusions

The observed high frequency of resistance among domestic and wild animals highlights the great zoonotic potential of circulating rapidly growing NTM strains, which could lead to challenges in the treatment of such infections. Significant gaps and limitations exist in our knowledge about diseases caused by NTM. To address these gaps, an enhanced understanding of ecological niches and the mechanisms by which organisms from these niches acquire resistance is needed.

## Figures and Tables

**Table 1 microorganisms-11-02520-t001:** MIC Breakpoints for Interpreting Susceptibility Tests of Rapidly Growing Nontuberculous Mycobacteria.

Antimicrobial	MIC, µg/mL		
Amikacin ^1^	≤16	32	≥64
Cefoxitin ^1^	≤16	32–64	≥128
Ciprofloxacin ^1^	≤1	2	≥4
Clarithromycin ^1^	≤2	4	≥8
Doxycycline ^1^	≤1	2–4	≥8
Imipenem ^1^	≤4	8–16	≥32
Linezolid ^1^	≤8	16	≥32
Moxifloxacin ^1^	≤1	2	≥4
Trimethoprim–sulfamethoxazole ^1^	≤2/38	-	≥4/76
Tigecycline ^1^	-	-	-
Tobramycin ^1^	≤2	4	≥8
Cefepime ^3^	≤8	16	≥32
Amoxicillin/clavulanic acid, ratio 2:1 ^2^	≤8/4	16/8	≥32/16
Ceftriaxone ^2^	≤8	16–32	≥64
Minocycline ^2^	≤1	2–4	≥8
**Interpretation**	**S**	**I**	**R**

I—intermediate; MIC—minimal inhibitory concentration; R—resistant; S—susceptible. ^1^ Antimicrobial agents and their breakpoints according to CLSI recommendations for testing RGM [19]. ^2^ Antimicrobial agents and their breakpoints according to CLSI recommendations for testing *Nocardia* spp. and other aerobic *Actinomycetes* [19]. ^3^ Antimicrobial agents and their breakpoints according to CLSI recommendations for testing *Nocardia* spp. and other aerobic *Actinomycetes* [23].

**Table 2 microorganisms-11-02520-t002:** Numbers of resistant rapidly growing NTM isolates to the indicated antibiotic.

NTM Species	No. Isolates Tested	No. Resistant Isolates													
		AMI	AUG2	FEP	FOX	AXO	CIP	CLA	DOX	IMI	LZD	MIN	MXF	TOB	SXT
*M. arupense*	2	0	0	0	0	0	0	0	0	0	0	0	0	0	0
*M. chitae*	6	0	2	1	0	0	0	2	0	0	0	0	0	0	0
*M. elephantis*	1	0	0	0	0	0	0	0	0	1	0	0	0	0	0
*M. fortuitum*	27	0	18	21	1	22	0	10	5	0 *	0	3	0	6	0
*M. neoaurum*	25	0	1	17	2	6	0	1	0	0	1	0	0	1	1
*M. peregrinum*	1	0	1	1	0	1	0	0	0	NR	0	0	0	1	0
*M. phocaicum*	7	0	2	1	0	1	0	0	0	0 **	0	0	0	1	0
*M. porcinum*	2	0	1	2	0	2	0	2	2	0	0	1	0	2	0
*M. pulveris*	1	0	0	0	0	0	0	0	0	0	0	0	0	0	0
*M. septicum*	1	0	0	0	0	0	0	0	0	0	0	0	0	0	0
*M. vaccae*	14	0	2	7	0	9	1	4	2	2	0	2	0	2	0

AMI—amikacin; AUG2—amoxicillin/clavulanic acid, ratio 2:1; AXO—ceftriaxone; CIP—ciprofloxacin; CLA—clarithromycin; DOX—doxycycline; FEP—cefepime; FOX—cefoxitin; IMI—imipenem; LZD—linezolid; MIN—minocycline; MXF—moxifloxacin; NR—not reported; SXT—trimethoprim-sulfamethoxazole; TOB—tobramycin. * The results for six isolates were not reported due to antibiotic instability. ** The results for four isolates were not reported due to antibiotic instability.

**Table 3 microorganisms-11-02520-t003:** Distribution of resistant rapidly growing NTM strains by antibiotic classes.

Antibiotic Group	Antibiotic	No. Isolates Tested	S	I	R	Percentage of Resistant Strains (%)
Aminoglycosides	AMI	87	86	1	0	**0**
	TOB	87	68	6	13	**14.9**
Penicillins	AUG2	87	46	14	27	**31.0**
Cephalosporins	FEP	87	25	12	50	**57.5**
	FOX	87	74	10	3	**3.4**
	AXO	87	23	23	41	**47.1**
Carbapenems	IMI	76 *	50	23	3	**3.9**
Fluoroquinolones	CIP	87	83	3	1	**1.1**
	MXF	87	87	0	0	**0**
Macrolides	CLA	87	64	3	20	**23.0**
Tetracyclines	DOX	87	61	17	9	**10.3**
	MIN	87	65	16	6	**6.9**
Oxazolidinones	LZD	87	84	2	1	**1.1**
Trimethoprim-sulfamethoxazole	SXT	87	86	0	1	**1.1**

AMI—amikacin; AUG2—amoxicillin/clavulanic acid, ratio 2:1; AXO—ceftriaxone; CIP—ciprofloxacin; CLA—clarithromycin; DOX—doxycycline; FEP—cefepime; FOX—cefoxitin; I—intermediate susceptible; IMI—imipenem; LZD—linezolid; MIN—minocycline; MXF—moxifloxacin; R—resistant; S—susceptible; SXT—trimethoprim–sulfamethoxazole; TOB—tobramycin. * The results for 11 isolates were not reported due to antibiotic instability.

**Table 4 microorganisms-11-02520-t004:** Multidrug-resistance among the 87 rapidly growing NTM isolates.

Number of Antibiotics to which Isolate is Resistant	No. Resistant Isolates (%)
2	15 (17.2)
3	6 (6.9)
4	5 (5.7)
5	5 (5.7)
6	1 (1.1)
**Total**	**32 (36.8)**

**Table 5 microorganisms-11-02520-t005:** The number of resistant strains to one or more antibiotics between different rapidly growing NTM species.

Number of Antibiotics to Which Isolate Is Resistant		*M. fortuitum*	*M. neoaurum*	*M. vaccae*	All Other NTM Species	No. Resistant Isolates
0	N	2	6	2	10	20
	%	7.41	24.00	47.62	14.29	22.99
1	N	2	12	4	6	24
	%	7.41	48.00	28.57	28.57	27.59
2	N	5	5	5	1	16
	%	18.52	20.00	4.76	35.71	18.39
3	N	8	1	0	1	10
	%	29.63	4.00	4.76	0.00	11.49
4 and more	N	10	1	3	3	17
	%	37.04	4.00	14.29	21.43	19.54
Total no. isolates tested	N	27	25	14	21	87
	%	100	100	100	100	100

No statistically significant differences were observed in the resistance patterns of rapidly growing mycobacteria strains isolated from domestic versus wild animals (*p* = 0.834).

## Data Availability

Data are available upon reasonable request.

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
