# Peer review of "Antimicrobial Resistance in Rapidly Growing Nontuberculous Mycobacteria among Domestic and Wild Animals Emphasizing the Zoonotic Potential"

_microorganisms, 2023, doi:10.3390/microorganisms11102520_

Round 1

Reviewer 1 Report

Dear Authors,

your manuscript reports a very interesting study. The research has been carried out with proper methodology and the results have been well presented and discussed. However, you could increase the quality of the paper adding a paragraph at the end of the discussion about the role of both wild and domestic animals in the spreading of NTM.

A little correction: pag 3, name of gene in italic form.

Author Response

Response to Reviewer 1 Comments

1. Summary

2. Point-by-point response to Comments and Suggestions for Authors

Comments 1: However, you could increase the quality of the paper adding a paragraph at the end of the discussion about the role of both wild and domestic animals in the spreading of NTM.

Response 1: Thank you for pointing this out. We agree with this comment. Therefore, we wrote a paragraph at the end of the discussion, which you can find in the revised manuscript.

Comments 2: A little correction: pag 3, name of gene in italic form.

Response 2: Agree. We have, accordingly, modified it.

Reviewer 2 Report

microorganisms-2611464 review v1

This manuscript provides valuable insights into NTM strains from domestic and wild animals and their resistance patterns. Although this manuscript emphasized the resistance of NTM strains, NTM are often considered as resistance to several antimicrobials. This natural resistance of NTM should be mentioned first and then discuss newly identified resistance which can disrupt the treatment in the clinical cases. Also, the meaning of the multidrug resistant should be clarified and then discussed. In addition to that, transmission of antimicrobial resistance of NTM strains should be discussed with relevant references.

For more comprehensive understanding of NTM characteristics, these should be clarified.

Please check these comments below:

Abstract: Please indicate how fast the NTM strains could grow (within X days). It could be also beneficial to readers if the abstract could provide more specific numbers for the antimicrobial resistance (%).

Line 21: minocycline and linezolid was substantially less prevalent. -> resistance to minocycline and linezolid was notably lower.

Line 23-24: species M. fortuitum, M. neoaurum, M. vaccae and M. porcinum, while the other species were in the -> species: M. fortuitum, M. neoaurum, M. vaccae, and M. porcinum. Meanwhile, other species displayed a …

Line 24-25: highest percentage sensitive. -> higher sensitivity rate.

Line 25: No differences in resistance among domestic and wild animals were -> No significant resistance differences between domestic and wild animals were

Line 26-27: the great zoonotic potential of -> the significant zoonotic potential posed by

Line 27: treatment of such -> treatment of these

Line 35-36: common called non-tuberculous mycobacteria- -> commonly referred to as non-tuberculous mycobacteria

Line 37-38: NTM species, also called ecological mycobacteria, -> Often termed ecological mycobacteria, NTM species

Line 38-39: divided into rapidly and slowly-growing species -> categorized as rapidly-growing and slowly-growing species

Line 43-44: their treatment is much more challenging than treatment … -> their management poses greater challenges than the treatment of

Line 50-51: caused by RGM have been reported from the most developed areas -> RGM-caused infections are reported predominantly from developed regions

Line 51-53: form of skin, soft tissue, bone and pulmonary disease [4]. It is a worrying fact -> manifesting as skin, soft tissue, bone, and pulmonary diseases [4]. The concerning aspect is

Line 63-64: in order to see the possible zoonotic potential and the possible role of animals -> to assess potential zoonotic implications and to understand the potential role of animals"

Line 73-75: The samples were collected in the case of a positive reaction to the tuberculin test and findings of pathological changes during slaughter -> Samples were obtained when there was a positive tuberculin test reaction or observable pathological changes suggesting tuberculosis during slaughter

Line 98-99: "examined the test plates for the first time after 48 hours." -> "first inspected the test plates after 48 hours."

Line 111-112: we repeated the test with incubation for no longer than three days. -> the test was repeated, with incubation limited to three days.

Line 117-118:  (MIC) values and interpretation of the results, following the guidelines described by the CLSI [16, 17] -> (MIC) values, interpreting results according to CLSI guidelines [16, 17]

Line 127: Table 1. Breakpoints of the minimum inhibitory concentration (MIC) for interpreting susceptibility tests of rapidly-growing nontuberculous mycobacteria -> Table 1: MIC Breakpoints for Interpreting Susceptibility Tests of Rapidly-growing Nontuberculous Mycobacteria

Table 1: Since this study used fast growing NTM, it will be better to include the timepoint for the MIC reading and also the reference information in the table footnote.

Line 143-144: The significance of the differences is shown by the P-value, whereby a P-value of less than 0.05 was considered statistically significant -> Differences were deemed statistically significant with a P-value less than 0.05.

Line 149: M. elephantis (1 isolates) -> M. elephantis (1 isolate)

Line 162: M. arupense, M. pulveris and M. septicum showed sensitivity to all tested antimicrobials. -> Isolates of M. arupense, M. pulveris, and M. septicum were sensitive to all tested antimicrobials.

Line 165: Table S1. -> Table 1.

Line 169, Table 2, 177, 196, Table 3: AUG2 -> AMO

Abbreviations for antimicrobials can be found here (https://journals.asm.org/abbreviations-conventions).

Table 3: Please correct the numbers considering the significant figures:  31 -> 31.0, 23 -> 23.0

Table 4: (% of 87) -> (%)

The term ‘Multidrug resistance (MDR)’ should be used for the two or more classes of drugs. Please include how many of them were MDR. It can be described in the same table or you can change the table to show the number of resistant antimicrobial classes.

Line 224-225: Observed differences in sensitivity to tested antibiotics between different species of rapidly-growing mycobacteria were statistically significant (p<0.001) -> There were statistically significant differences in antibiotic sensitivity among the species of rapidly-growing mycobacteria (p<0.001).

Line 232-233: Observed differences in the occurrence of resistant strains of rapidly-growing mycobacteria isolated from domestic and wild animals were not statistically significant (P=0.834). -> No statistically significant differences were observed in the resistance patterns of rapidly-growing mycobacteria strains isolated from domestic versus wild animals (P=0.834).

Line 265: Our isolates were resistant to -> Our M. chitae isolates exhibited resistance to

Line 268: described as a cause -> known to cause

Line 282: antimicrobial drugs for the treatment of infections -> recommended antimicrobials for treating infections

Line 302: treatment of such infections in humans. -> treatment of infections in humans.

Line 322: domestic and nine from wild animals. -> domestic animals and nine from wild animals.

Line 324: that have not been in contact -> that likely have not been exposed

Line 326: diseases caused by rapidly-growing NTMs. -> the diseases caused by rapidly-growing NTM species.

Line 330: The established significant frequency -> The observed high frequency

Line 333: There are many weaknesses and gaps -> There exist significant gaps and limitations

Line 334: In order to clarify these things, -> To address these gaps,

Line 335: a better understanding of ecological niches -> an enhanced understanding of ecological niches

Moderate editing of English language required.

Round 2

Reviewer 2 Report

The manuscript was revised well and will provide valuable information for readers.